# Transcribed Ultraconserved Regions Are Associated with Clinicopathological Features in Breast Cancer

**DOI:** 10.3390/biom12020214

**Published:** 2022-01-26

**Authors:** Erika Pereira Zambalde, Douglas Adamoski, Daniela Fiori Gradia, Iris Rabinovich, Ana Carolina Rodrigues, Cristina Ivan, Enilze M. S. F. Ribeiro, George Adrian Calin, Jaqueline Carvalho de Oliveira

**Affiliations:** 1Laboratory of Human Cytogenetics and Oncogenetics, Department of Genetics, Universidade Federal do Paraná, Curitiba 81531-980, PR, Brazil; erikazambaldi@gmail.com (E.P.Z.); danielagradia@ufpr.br (D.F.G.); ana.acr.rodrigues@gmail.com (A.C.R.); enilzeribeiro@gmail.com (E.M.S.F.R.); 2Brazilian Biosciences National Laboratory (LNBio), Brazilian Center for Research in Energy and Materials (CNPEM), Campinas 13083-970, SP, Brazil; douglas.adamoski@gmail.com; 3Hospital Nossa Senhora das Graças, Centro de Doenças da Mama, Curitiba 80810-040, PR, Brazil; iris_rbv@yahoo.com.br; 4Translational Molecular Pathology Department, The University of Texas MD Anderson Cancer Center, TX 77230, USA; CIvan@mdanderson.org (C.I.); gcalin@mdanderson.org (G.A.C.); 5Center for RNA Interference and Non-Coding RNAs, The University of Texas MD Anderson Cancer Center, Houston, TX 77230, USA

**Keywords:** T-UCRs, breast cancer (BC), lncRNAs

## Abstract

Ultraconserved regions (UCRs) are 481 genome segments, with length longer than 200 bp, that are 100% conserved among humans, mice, and rats. The majority of UCRs are transcriptionally active (T-UCRs) as many of them produce non-coding RNAs. In a previous study, we evaluated the expression level of T-UCRs in breast cancer (BC) patients and found that 63% of transcripts correlated with some clinical and/or molecular parameter of BC. In this study, we delved into the expression levels of 12 T-UCRs and correlated them with clinicopathological parameters, immunohistochemical markers, and overall survival in two breast cancer cohorts: TCGA and Brazilian patients. We found that uc.268 is more expressed in TCGA patients under 40 years of age, associated with progesterone receptor (PR) and estrogen receptor (ER), and its high expression is found in luminal A. Lower uc.84 and uc.376 were respectively observed in metastatic and stage IV tumors associated with good prognostic in luminal B. Moreover, uc.84 was only related to the HER2+, while uc.376 was related to ER+ and PR+, and HER2+. A panel composed of uc.147, uc.271, and uc.427 distinguished luminal A from triple negative patients with an AUC of 0.9531 (sensitivity 92.19% and specificity 86.76%). These results highlight the potential role of T-UCRs in BC and provide insights into the potential application of T-UCRs as biomarkers.

## 1. Introduction

Human genome sequencing demonstrated that only 2% of DNA are protein-coding genes. The remaining 98% constitutes what was previously called the “dark matter” of molecular biology and includes hidden information mainly responsible for phenotype regulation. Dark matter has become a new field to explore [1]. Nowadays, it is proven that this region can be transcribed into distinct RNAs that have an impact on gene regulation [2].

Those RNAs were named non-coding RNAs (ncRNAs), which include a variety of molecules and are subdivided into two major groups: miRNAs (with a sequence less than 200 bp) and long non-coding RNA (with a sequence of more than 200 bp) [3].

Ultraconserved regions (UCRs) are regions of conserved DNA segments, which have 100% identity among humans, rats, and mice [4]. Most of these regions are transcriptionally active (T-UCRs), and 80% are transcribed from genomic elements located in both intra- and intergenic regions, representing a class of long-non-coding RNAs [5]. The extreme conservation of T-UCRs in non-coding regions and the significant number of deregulated T-UCRs associated with human diseases may be a sign of their importance in physiologic and pathophysiologic processes [6,7]. Most UCRs are known to be transcribed ultraconserved regions (T-UCRs) in normal and malignant tissues [6]. In addition, T-UCRs are more frequently found in cancer-associated genomic regions (CAGRs), and the differential expression of T-UCRs has been described in several types of cancer [8]. Several studies have identified distinct signatures in human carcinomas, such as hepatocellular (HCC) [9,10,11], neuroblastoma (NB) [5], chronic lymphocytic leukemia (CLL) [6], prostate cancer (PC) [12], and bladder cancer (BlaC) [13]. T-UCRs expression profiles can be a useful tool to differentiate human cancer types and improve their diagnosis and prognosis. In breast cancer (BC), however, the utility of T-UCRs as a biomarker is still underexplored.

BC is the most commonly diagnosed type of cancer globally and the leading cause of cancer death among women worldwide [14]. In addition, BC is a heterogeneous disease with a range of clinical and molecular characteristics [15]. Based on immunohistochemical analysis, BC is classified into four main groups: luminal A, luminal B, HER2 (Human Epidermal Growth Factor Receptor 2) enriched, and triple-negative [16,17]. This classification is essential for prognosis and treatment decisions. The triple-negative is the most challenging group to treat, but new targeted therapies are becoming available, e.g., PARP inhibitors in BRCA mutated patients [18]. Monoclonal antibody therapy has excellent results in HER2 enriched patients [19]. Luminal A and luminal B patients can benefit from hormone therapies. Moreover, BC can also be categorized according the expression profile of 50 genes using partition around medoids, a method called PAM50. In this classification BC is stratified into five groups: luminal A, luminal B, HER2 enriched, basal, and normal-like [20,21].

Despite the BC general classification system and the existing guidelines for the stratification of patients inside each group, the wide range of diversity observed in each subtype demands additional molecular markers to offer personalized therapy and improve BC survival rates [22,23]. In addition, only four T-UCRs have been strongly associated with BC: uc.38, downregulated BC patients [24], uc.63, upregulated in luminal A patients associated with poor prognosis [25], lnc-uc.147, associated with poor prognosis in luminal A [26], and uc.51, associated to metastasis [27].

In a previous study from our group, we evaluated the expression level of all 481 T-UCRs in TCGA (The Cancer Genome Atlas) BC patients and found that 63% of transcripts correlated with some clinical and/or molecular parameter of BC [26]. Including a T-UCR transcriptional portrait, 12 T-UCRs (uc.84, uc.138, uc.147, uc.193, uc.268, uc.271, uc.311, uc.376, uc.378, uc.427, uc.456, uc.475) were suggested to have great potential as biomarkers in BC, but these molecules were not further investigated [26]. The present study aimed to evaluate the 12 T-UCRs in the TCGA data and the Brazilian cohort. We also associated the T-UCRs with several parameters such as metastasis, age, overall survival, and positivity of receptors. All the evidence found suggests that T-UCRs should be explored as a prognostic marker and target for therapies in BC.

## 2. Materials and Methods

### 2.1. Breast Cancer Patients

This study was approved by the Ethical Committee in Research of the Federal University of Paraná (UFPR) (CONEP: 19870319.3.0000.0102), and all individuals signed a written informed consent form. Tumor samples from 102 breast cancer patients (30 luminal A, 30 luminal B, 9 HER2-enriched, and 33 triple-negative) were collected at Hospital Nossa Senhora da Graças, Curitiba, Paraná, Brazil. Tissue samples were obtained from fresh surgical specimens later frozen in RNA later and stored at −80 °C. The subtypes of all samples were immunohistochemically confirmed. Patients’ clinicopathological characteristics are listed in Table 1 and Appendix A.

### 2.2. RNA Extraction, cDNA Synthesis, and RT-qPCR

RNA from tissues was extracted using the Quick RNA Miniprep Kit (Zymo Research, Irvine, CA, USA). All RNAs were treated with DNase (Ambion, Austin, TX, USA). RNA quality and concentration were assessed using the nanodrop ND-1000 instrument (NanoDrop Technologies, Thermo Scientific, Wilmington, DE, USA) and Bioanalyzer (Agilent, Santa Clara, CA, USA). cDNA was synthesized using the SuperScript III cDNA kit (Invitrogen, Waltham, MA, USA), and diluted cDNA was used for RT-PCR analysis using iQ SYBR Green Supermix (Bio-Rad, Richmond, CA, USA) with the appropriate primers (Appendix A). The MCF-7 cell line was used as a calibrator sample. The 2^−ΔΔCt^ method was used to calculate the relative abundance of RNA genes compared with two of the following genes: the TBP and u6 expression. All experiments were performed with a RT-control to identify DNA contamination.

### 2.3. TCGA Data

The statistical analyses for the TCGA data were performed in R software (version 3.4.1, R, Vienna, Austria) (http:///www.r-project.org/ (accessed on 14 January 2022)), and the statistical significance was defined as a *p*-value less than 0.05. The ultraconserved elements’ (UCR) sequences in the human genome, their genomic coordinates in GRCh34/hg16, as well their type (exonic, nonexonic) and the closest gene were retrieved from https://users.soe.ucsc.edu/~jill/ultra.html (accessed on 17 November 2021) [4]. We used UCSC Genome Browser LiftOver to lift the coordinates from GRCh34/hg16 to GRCh37/hg19 assembly. The expression of the ultraconserved regions, quantified as RPKM (reads per kilobase of transcript per million mapped reads) in primary tumors, were retrieved from TANRIC (https://ibl.mdanderson.org/tanric/_design/basic/main.html (accessed on 10 January 2022)) [28]. We downloaded clinical information for the TCGA patients with invasive breast carcinoma (BRCA) from the cbioPortal (http://www.cbioportal.org/(accessed on 14 January 2022)), (cohort: TCGA provisional). Hormonal marker status for Estrogen Receptor (ER), Progesterone Receptor (PR), and Human Epidermal growth factor Receptor 2 (HER2) were defined from TCGA’s IHC data. The PAM50 subtype of the samples was retrieve from [29]. Updated TCGA primary end point data as survival status and follow-up time, were obtained from [30]. For overall survival (OS): overall survival event included death from any cause. For progression-free interval (PFI), event considered patients having new tumor event whether it was a progression of disease, local recurrence, distant metastasis, new primary tumors all sites, or died with the cancer with or without new tumor event.

We obtained 827 cases with UC expression and clinical information. From the Genomic Data Commons Data Portal, we downloaded FPKM (fragments per kilobase of transcript per million mapped reads) files to retrieve gene expression quantification data for the host genes of the ultraconserved regions in the same tumors. The log2-transformation was applied to the mRNA data.

### 2.4. Statistical Analysis

The association between the variables analyzed and T-UCRs expression levels was determined by non-parametric test (Mann–Whitney preceded by Kruskal–Wallis test when multiple comparisons were performed). The relationship between overall survival and covariates (mRNA expression levels and clinical parameters, such as age and stage) was examined through a Cox proportional hazard model. A multivariate Cox proportional hazard model was fitted, including the clinical parameters and mRNA expression significance in the univariate analysis. These analyses were conducted for the breast cancer cohorts and separately on each subtype as defined by PAM50 or immunohistochemistry. The adjusted *p*-value of <0.05 was considered significant. Receiver operating characteristic (ROC) curves were calculated based on RQ values. For combined ROC curves, a binary logistic regression was calculated using IBM SPSS Statistics (IBM SPSS Statistics Inc., Armonk, NY, USA). We use the software IBM SPSS Statistics to perform the binary logistic. In this analysis, we used the expression data already obtained. We selected as dependent variable the tumor samples (if we are analyzing tumor versus non-tumor) or luminal A samples (if we are analyzing luminal A versus TNBC). Subsequently, we selected the expression levels of the T-UCRs as the variable that we wish to combine into the covariates deriving Cox regression β values for each T-UCR, which were used as weighting factors to derive an overall score of combined T-UCRs expression for each patient. This score were used as the entry for ROC analysis. The true positive rate (sensitivity) versus the false positive rate (1-specificity) were plotted at various threshold settings, and the optimal cutoff threshold was calculated using Youden’s index (highest sensibility plus specificity).

## 3. Results

### 3.1. Differential Expression of T-UCRs in BC Samples

All T-UCRs analyzed had expression detected in more than 80% of the samples analyzed in both cohorts. Analysis of the differential expression of T-UCRs uc.84, uc.138, uc.147, uc.193, uc.268, uc.271, uc.311, uc.376, uc.378, uc.427, uc.456, and uc.475 in TCGA BC tumor samples (*n* = 827) versus non-tumor (*n* = 105) demonstrated that 10 T-UCRs (uc.138, uc.147, uc.193, uc.268, uc.271, uc.311, uc.376, uc.378, uc.427, uc.456) had differential expression (Figure 1). The T-UCRs uc.138, uc.311, uc.376, and uc.456 were down expressed while uc.147, uc.193, uc.268, uc.271, uc.378, and uc.427 were highly expressed in the tumor samples.

The diagnostic potential of T-UCRs was analysed through receiver operating characteristic (ROC) curves (Figure 2). The T-UCRs uc.147 and uc.271 had the highest area under the curve (AUC) of 0.64 (uc.147: sensitivity of 61.32% and specificity of 63.81%; uc.271: sensitivity of 60.67% and specificity of 68.60%), followed by uc.268 and uc.427, both with an AUC of 0.60 (uc.268: sensitivity of 56.47% and specificity of 68.57% and uc.475: sensitivity of 49.74% and specificity of 74.47%) (Figure 2). By combining these four T-UCRs, AUC increases to 0.74, with 65.42% sensitivity and 73.33% specificity (Figure 2E). Interestingly, when we create a panel combining all 12 T-UCRs, AUC rises to 0.93 along with higher sensitivity and specificity (85.37% and 85.71%, respectively), indicating a great association of these T-UCRs and breast cancer (Figure 2F). Another combination of seven T-UCRs (uc.138, uc.147, uc.268, uc.271, uc.311, uc.427, and uc.475) also demonstrated high AUC (0.92) and good sensitivity (83.09%) and specificity (87.62%), which indicates that a panel of these seven T-UCRs could be enough for the BC diagnostic (Figure 2G). Other combinations were also tested, but those aforementioned represent the best combinations of the T-UCRs associated with BC.

### 3.2. Differential Expression of T-UCRs in BC Subtypes

Classifying subtypes in BC is essential for prognosis and treatment decisions. All 12 T-UCRs from TCGA cohort (Figure 3) and two T-UCRs (uc.147 and uc.193) in the Brazilian (102) cohort presented differential expression among subtypes.

We also tested the potential of T-UCRs to discriminate the subtypes luminal A (better prognostic) and triple-negative (worse prognostic). We noted that uc.147 has better potential do differentiate both subtypes in both cohorts, TCGA and Brazilian, with an AUC of 0.822 (sensitivity of 85.94% and specificity of 66.76%) and 0.815 (sensitivity of 88.89% and specificity of 66.67%) respectively (Figure 4A,B). For the TCGA cohort, two more T-UCRs were good markers: uc.271 (AUC: 0.865; sensitivity of 85.16% and specificity of 78.85%) and uc.427 (AUC: 0.875; sensitivity of 85.16% and specificity of 78.85) (Figure 4C,D).

Interestingly, the discriminatory accuracy of the ROC curve is improves combined with the expression levels of uc.147, uc.271, and uc.427, which also presented good AUC values. The combined AUC of these T-UCRs improved to 0.9531 (Figure 4E), as well as the sensitivity and specificity values (92.19% and 86.76%, respectively). Together, these results indicate that the selected T-UCRs, either individually or combined, can distinguish BC between the luminal A and TNBC (triple negative breast cancer) subtypes with high sensitivity and specificity.

### 3.3. Association of T-UCRs Expression with Clinicopathological Parameters of BC

Immunohistochemical analysis was used to stratify BC and therapy decisions. Most T-UCRs analyzed in TCGA have differential expression according to progesterone and estrogen receptors. Out of the 12 T-UCRs analyzed, 10 (uc.147, uc.193, uc.268, uc.271, uc.311, uc.376, uc.378, uc.427, uc.456, uc.475) showed association with the progesterone receptor (Appendix A), 11 (uc.138, uc.147, uc.193, uc.268, uc.271, uc.311, uc.376, uc.378, uc.427, uc.456, uc.475) with the estrogen receptor (Appendix A), and four (uc.84, uc.138, uc.376 and uc.475) with the HER2 status (Appendix A).

In TCGA data, uc.268 is more expressed in diagnosed patients who are under 40 years old (Figure 5A). A lower expression of uc.84 was observed in patients with metastasis (Figure 5B). In addition, stage IV tumors showed a lower expression of uc.376 (Figure 5C). These results suggest that uc.268 could be helpful to identify risk for earlier BC, while uc.84 and uc.376 could indicate less aggressive cancer. In the Brazilian cohort, no association was found, possibly due to a small number of samples.

### 3.4. T-UCRs Expression in BC Correlates with Patients’ Overall Survival

Using univariate analysis, we noted that uc.193 is associated with overall survival (OS) in breast cancer. Using a multivariate analysis, we demonstrated that high uc.193 expression is an independent prognostic marker of poor survival for BC (HR = 1.75, IC 95%: 1.21–2.52, *p* < 0.01) (Table 2). We also analyzed different subtypes and for the hormonal and HER2 receptors.

In luminal A, a subtype usually associated with better prognosis, the expression of six T-UCRs (uc.84, uc.147, uc.193, uc.268, uc.456 and uc.475) was associated with survival time in uni- and multivariate analysis, highlighting uc.193 with a 6.54 HR (Appendix A).

Another five T-UCRs (uc.138, uc.147, uc.311, uc.376, and uc.456) were indicated as luminal B subtype correlated factors, an intermediate subtype in BC. In the HER2 subtype, which is an aggressive subtype but has targeted therapy available, the uc.193 was shown to associated to this subtype (Table 3). No T-UCRs analyzed were associated with overall survival in triple-negative.

These results demonstrated a relation between the T-UCRs and the survival rate in different subtypes, evidencing that the T-UCRs could also be used as a prognostic marker.

### 3.5. T-UCRs Expression in BC Correlates with Patients’ Progression Free Interval (PFI)

In addition to overall survival, we decided to evaluate the patients’ progression free interval (PFI) that assesses if the person had a continuation of the cancer, metastasis, or locoregional recurrence and is not polluted with someone who died of other causes than cancer. This analysis was performed with updated TCGA data [30].

Contradictory to overall survival, none of T-UCRs were associated with PFI in general breast cancer when we did the univariate analysis (Table 4). However, when we stratified the TCGA cohort, we found a subtype-specific association between PFI and T-UCRs expression (Table 5).

In luminal A, five T-UCRs (uc.84, uc.147, uc.268, uc.456 and uc.475) were associated to PFI in the univariate analysis and four of them (uc.84, uc.147, uc.268, and uc.475) also had an association with PFI in multivariate analysis. The difference here is for the uc.193 that had a greater association with overall survival, but no association with PFI.

According to uni- and multivariate analyses, three T-UCRs (uc.84, uc.147 and uc.311) were correlated to the PFI in luminal B subtype. Hence, uc.84 was associated only with PFI, but not with overall survival. In addition, uc.147 and uc.378 were associated with PFI in the HER2 subtype, but only uc.147 continued to have a significant correlation in the multivariate analysis, indicating the high correlation of these T-UCR with the HER2 subtype (Table 5). These results are more specific and provide a more detailed analysis concerning the association of T-UCRs with the survival rate in different subtypes, revealing breast tumor heterogeneity and the relevance of T-UCRs in specific contexts.

## 4. Discussion

Breast cancer is a heterogeneous disease with many clinical and molecular characteristics [15]. Despite the BC general classification system, enhancements to intrinsic subtype classifications need to be proposed to improve prognosis, recurrence risk, survival, or treatment options [31]. Moreover, tumor characteristics can vary according to the population [32].

LncRNAs hold a great promise to be used as biomarkers for diagnosis and prognosis in breast cancer. Studies are already underway to have them developed as the basis of next-generation therapeutics for various cancers [33]. Nonetheless, lncRNAs derived from the T-UCRs are underexplored as potential BC biomarkers. In a previous study, we analyzed all 481 T-UCRs in BC TCGA patients, showing that 63% of transcripts correlated with some clinical and/or molecular parameter of BC.

In the present more in depth analysis, we evaluated 12 T-UCRs according to molecular/clinical parameters in two distinct cohorts: TCGA cohort (827) and Brazilian patients (102).

Uc.84 was associated with PAM50 subtypes with higher expression in luminal A patients. Additionally, it was found to be less expressed in HER-2 positive patients and patients with metastasis events. These results associated uc.84 with less aggressive tumors. However, this may be a reflection of subtype association. Usually, luminal patients presents less metastasis compared to more aggressive subtypes. This idea is supported by the survival analysis only considering luminal A and B patients. In this group, the high expression of uc.84 patients was associated with worse survival time and, in this type of cell, may also be involved in more aggressive pathways. In a recent study, authors showed that a correlation of uc.84 with mir-221 is associated with the regulation of the *CDKND1* and plays pivotal role in the cell cycle in breast cancer cell lines [34].

On the other hand, uc.138 is under-expressed in tumorous tissues compared to non-tumorous ones. Uc.138 is less expressed in PR and HER-2 positive samples when associated with subtypes. In luminal B patients, high expression is associated with better survival, but this association is not seen in PFI. Uc.138 was previously associated with tumors, but in colon cancer [35,36]. In contrast with our observation in breast cancer, uc.138 is overexpressed and associated with a poor prognosis in colon cancer cases [36]. In colon cells, the transcript containing uc.138 is nuclear and influences cell proliferation by changing the expression of G2/M-related cell cycle regulators.

Uc.147 was previously highlighted by our group as an important lncRNA in breast cancer [26]. The silencing of uc.147 in luminal BC cell lines increases apoptosis, arrests cell cycle, and decreases colony formation and cell viability [26]. These experiments indicated that this T-UCR acts as an oncogene in BC cell lines. Therefore, here, we reinforce the importance of the lnc-uc.147, since this lncRNA was highly expressed in tumor versus non-tumor, in luminal subtypes, including association with ER/PR presence, and had an important association with poor survival in luminal and with PFI in luminal A/B and HER2 patients. The previous study combined with our results increases the potential of uc.147 as a prognostic marker in BC.

Uc.193 is also an important molecule that needs deeper analysis. This T-UCR is associated with tumor in general, molecular subtypes, and ER/PR presence. Uc.193 was associated with poor overall survival considering all BC patients, luminal A, and HER-2 subtypes. However, when we looked at the PFI analysis, the uc.193 did not demonstrate any association with survival, showing a potential confounder with types of death excluding cancer. We found that uc.268 was highly expressed in tumor cells and associated with molecular subtype and ER/PR presence. Additionally, uc.268 is more expressed in diagnosed patients under 40 years age. In luminal A patients, the high expression was associated with a poor survival rate, so apparently the expression of this region is more related to worse features.

Uc.271, uc.378, and uc.427 have a similar expression profile, as they are highly expressed in tumorous tissues compared to non-tumorous ones, associated with molecular subtypes, and are more expressed in PR/ER positive patients. On the other hand, uc.311 and uc.376 were less expressed in tumorous versus non-tumorous tissues, also associated with molecular subtypes and ER/PR positivity.

Uc.456 was hypoexpressed in tumor cells, associated with molecular subtypes, and its high expression was found in ER/PR tumors. Interestingly, in luminal A patients, uc.456 expression was associated with a poor survival rate in OS and PFI analysis. Uc.475 expression was different in molecular subtypes including ER/PR positive patients. In the luminal A group, it was associated with poor prognosis. Uc.475 is considered a hypoxia-induced, noncoding, ultraconserved transcript that is overexpressed in epithelial cancer types [37]. In colon cell lines, under hypoxic conditions, uc.475 knockdown decreased cell proliferation by G2/M arrest. In breast cancer, the role of this UCR is still under investigated.

An important aspect of the present study is that the diagnostic potential of T-UCRs was evaluated through individual and combined ROC curves. The combination of all 12 studied T-UCRs demonstrated high impact as a diagnostic marker for BC, with an AUC of 0.93. Another combination of seven T-UCRs (uc.138, uc.147, uc.268, uc.271, uc.311, uc.427, and uc.475) also demonstrated an AUC of 0.92. These results suggest an interesting panel able to differentiate tumorous versus non-tumorous tissue.

Subtype classification is a key determinant for patient outcome and therapeutic choices. In this study, the analysis of T-UCRs levels within two major BC groups, luminal A and TNBC, indicated that high expression levels of uc.147 were able to discriminate luminal A patients from TNBC patients with high accuracy (AUC of 0.822 and 0.815 for TCGA and Brazilian cohort, respectively). Interestingly, the discriminatory accuracy of the ROC curve improves when combined with the expression levels of uc.147, uc.271, and uc.427, which also presented good AUC values (0.9531).

This is the first time that the association of T-UCRs with important clinicopathological features was deeply evaluated in BC. More studies are needed to validate the T-UCRs as potential biomarkers in BC as well as other cancers. Nevertheless, this study opens a field to be explored inside the ncRNAs world.

## 5. Conclusions

We presented the T-UCRs uc.84, uc.138, uc.147, uc.193, uc.268, uc.271, uc.311, uc.376, uc.378, uc.427, uc.456, and uc.475, their association with different clinical and molecular parameters in BC, and their correlation of features in different cohorts. In addition, we demonstrated that T-UCRs could be used to stratify patients inside each subtype in breast cancer and help therapy decisions. Altogether, these results indicate the potential of T-UCRs as a prognostic marker in breast cancer.

## Figures and Tables

**Figure 1 biomolecules-12-00214-f001:**
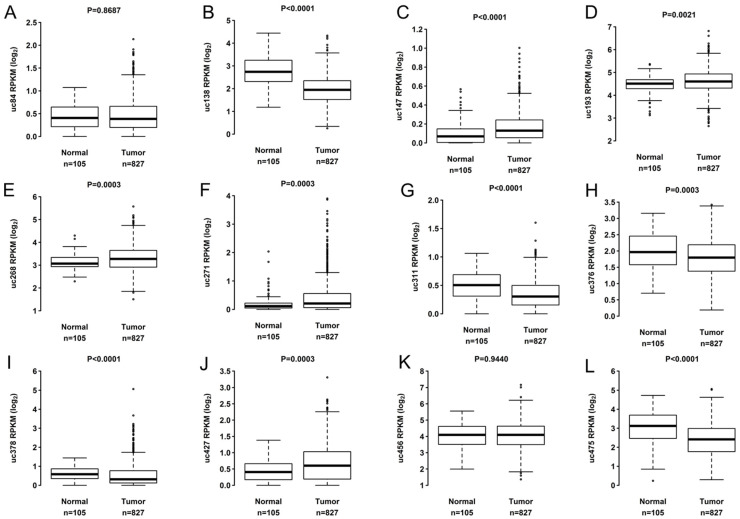
Expression levels of 12 T-UCRs. Comparison of expression levels of 12 T-UCRs in tumorous versus non- tumorous BC samples from TCGA. Differential expression level of (**A**) uc.84, (**B**) uc.138, (**C**) uc.147, (**D**) uc.193, (**E**) uc.268, (**F**) uc.271, (**G**) uc.311, (**H**) uc.376, (**I**) uc.378, (**J**) uc.427, (**K**) uc.456 and (**L**) uc.475 in tumorous versus non-tumorous. Each dot represents outlier patient.

**Figure 2 biomolecules-12-00214-f002:**
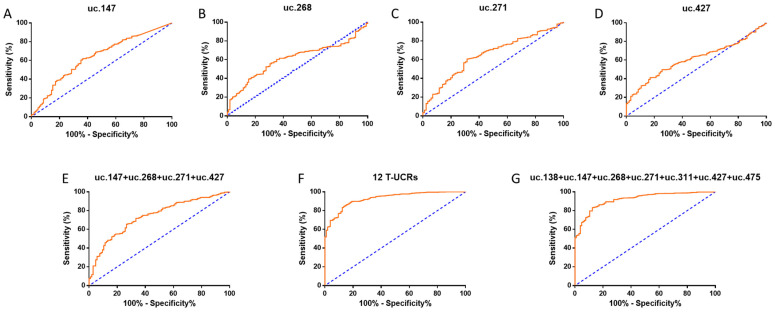
Individual and combined receiver operating characteristic (ROC) curves from T-UCRs, indicating diagnostic potential panels. (**A**–**D**) BC prognostics accuracy calculated for uc.147 TCGA (**A**), uc.268 TCGA (**B**), uc.271 TCGA (**C**), and uc.427 TCGA (**D**). (**E**–**G**) Combined ROC curves of uc.147 + uc.268 + uc.271 + uc.427 (**E**), all 12 T-UCRs (**F**), and uc.138 + uc.147 + uc.268 + uc.271 + 311 + uc.427 + uc.475 (**G**). ROC curves calculated based on relative quantification (RQ) values. All *p* values are under 0.05.

**Figure 3 biomolecules-12-00214-f003:**
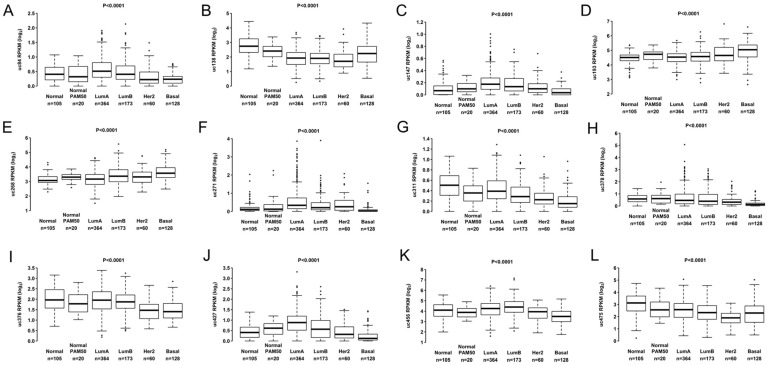
Expression levels of 12 T-UCRs in all breast cancer subtypes from TCGA patients: Normal, Normal PAM50, Luminal A (LumA), Luminal B (LumB), HER2 overexpression (HER2+), and Basal. (**A**) Differential expression level of uc.84, (**B**) uc.138, (**C**) uc.147, (**D**) uc.193, (**E**) uc.268, (**F**) uc.271, (**G**) uc.311, (**H**) uc.378, (**I**) uc.376, (**J**) uc.427, (**K**) uc.456 and (**L**) uc.475 according to PAM50 BC subtypes.

**Figure 4 biomolecules-12-00214-f004:**
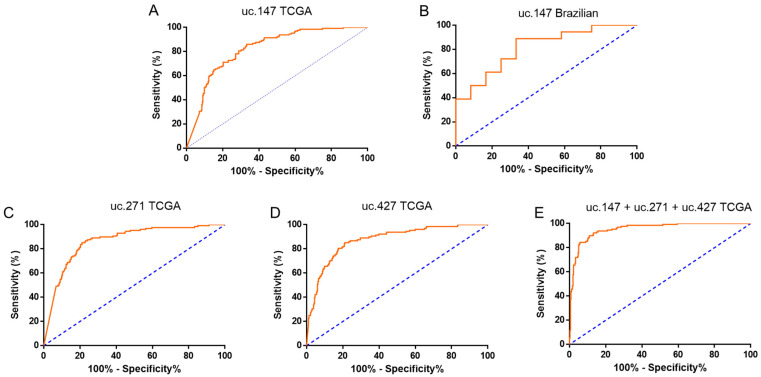
Individual and combined receiver operating characteristic (ROC) curves fromT-UCRs, indicating prognostic potential panels. BC prognostics accuracy calculated for uc.147 TCGA (**A**), uc.147 Brazilian cohort (**B**), uc.271 TCGA (**C**), and uc.427 TCGA (**D**). (**E**) BC prognostics accuracy calculated for uc.147 from Brazilian cohort. ROC curves calculated based on relative quantification (RQ) values.

**Figure 5 biomolecules-12-00214-f005:**
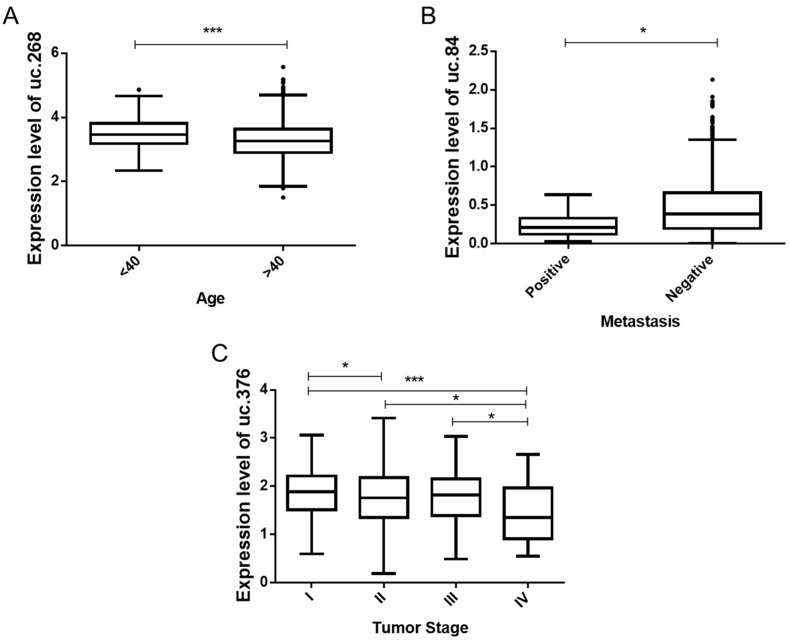
Comparison of T-UCRs expression to clinicopathological data from TCGA data. (**A**) Differential expression level of uc.268 between the patients’ ages (*p* < 0.001). (**B**) Significant down-relative expression of uc.84 patients with metastasis compared to patients negative for metastasis (*p* < 0.05). (**C**) Significant reduction in uc.376 (*p* < 0.05) between tumor stage I, II, III, and IV. * *p* < 0.05, and *** *p* < 0.001.

**Table 1 biomolecules-12-00214-t001:** Clinicopathological parameters of Brazilian breast cancer patients.

Parameters	Categorization	Sample Size	%
Age at diagnosis	Median 53 years		
	(range: 25–90)		
	≤53	51	50.0
	>53	48	47.0
	Unknown	3	2.9
Estrogen receptor status	Positive	59	57.8
	Negative	43	42.2
	Unknown	-	-
Progesterone receptor status	Positive	59	57.8
	Negative	43	42.2
	Unknown	-	-
HER2 status	Positive	55	53.9
	Negative	45	44.1
	Unknown	2	1.9
Proliferation index (Ki-67)	≤15%	33	32.4
	>15%	67	65.7
	Unknown	2	1.9
Immunohistochemical Subtype	Luminal A	30	29.4
	Luminal B	30	29.4
	HER2+	9	8.8
	Triple Negative	33	32.4
	Unknown	-	-
Lymph node status	Positive	30	29.4
	Negative	57	55.9
	Unknown	15	14.7
Distance Metastasis		11	10.7
Death		3	2.9

HER2: human epidermal growth factor receptor 2.

**Table 2 biomolecules-12-00214-t002:** Summary of univariate and multivariate Cox regression analysis of al T-UCRs in overall patients survival.

	Univariate Analysis					Multivariate Analysis			
	Variable	HR	Lower 0.95	Upper 0.95	*p*-Value	HR	Lower 0.95	Upper 0.95	*p*-Value
Overall Breast Cancer	Age	1.03	1.02	1.05	1.92 × 10^−5^ *	2.34	1.55	3.52	5.30 × 10^−5^ *
	Stage	2.34	1.55	3.52	5.30 × 10^−5^ *	1.04	1.02	1.05	5.9 × 10^−7^ *
	uc.84	1.19	0.74	1.92	0.46				
	uc.138	1.11	0.84	1.46	0.48				
	uc.147	2.49	0.79	7.83	0.12	1.75	1.21	2.52	0.0026 *
	uc.193	1.44	1.04	1.98	0.03 *				
	uc.268	1.19	0.86	1.64	0.29				
	uc.271	0.87	0.64	1.18	0.36				
	uc.311	0.64	0.31	1.34	0.23				
	uc.376	1.09	0.79	1.49	0.61				
	uc.378	1.03	0.77	1.39	0.83				
	uc.427	0.99	0.71	1.38	0.95				
	uc.456	1.18	0.95	1.48	0.14				
	uc.475	1.19	0.95	1.48	0.12				

* *p*-value < 0.05.

**Table 3 biomolecules-12-00214-t003:** Summary of univariate Cox regression analysis of T-UCRs in overall survival among subtypes.

		Univariate Analysis			
	Variable	HR	Lower 0.95	Upper 0.95	*p*-Value
Luminal A	Age	1.032	1.011	1.054	0.003 *
	Stage	1.726	0.916	3.254	0.091
	uc.84	2.202	1.170	4.145	0.014 *
	uc.138	1.555	0.980	2.466	0.061
	uc.147	8.042	1.891	34.207	0.005 *
	uc.193	2.629	1.320	5.236	0.006 *
	uc.268	1.761	1.029	3.016	0.039 *
	uc.271	0.764	0.499	1.168	0.214
	uc.311	0.949	0.330	2.727	0.922
	uc.376	1.454	0.870	2.429	0.153
	uc.378	1.217	0.840	1.763	0.298
	uc.427	1.161	0.693	1.945	0.571
	uc.456	1.604	1.085	2.372	0.018 *
	uc.475	1.588	1.089	2.317	0.016 *
Luminal B	Age	1.024	0.994	1.055	0.112
	Stage	2.108	0.959	4.632	0.063
	uc.84	0.220	0.042	1.140	0.071
	uc.138	0.530	0.288	0.973	0.041 *
	uc.147	0.002	0.000	0.080	0.001 *
	uc.193	1.811	0.867	3.784	0.114
	uc.268	0.688	0.354	1.336	0.269
	uc.271	2.006	0.862	4.668	0.106
	uc.311	0.101	0.011	0.933	0.043 *
	uc.376	0.412	0.191	0.888	0.024 *
	uc.378	0.911	0.444	1.869	0.799
	uc.427	0.569	0.247	1.312	0.186
	uc.456	0.624	0.391	0.997	0.049 *
	uc.475	1.035	0.616	1.738	0.897
HER2 positive	Age	1.073	1.030	1.119	0.001 *
	Stage	2.405	0.892	6.484	0.083
	uc.84	2.591	0.728	9.219	0.141
	uc.138	1.693	0.678	4.230	0.259
	uc.147	6.438	0.348	119.061	0.211
	uc.193	2.312	1.051	5.084	0.037 *
	uc.268	0.383	0.126	1.161	0.090
	uc.271	1.054	0.453	2.452	0.903
	uc.311	1.165	0.135	10.091	0.889
	uc.376	1.213	0.424	3.476	0.719
	uc.378 uc.427	0.403 0.478	0.091 0.132	1.784 1.736	0.231 0.262
	uc.456	1.266	0.600	2.674	0.535
	uc.475	1.440	0.707	2.933	0.315

* *p*-value < 0.05.

**Table 4 biomolecules-12-00214-t004:** Summary of univariate Cox regression analysis of al T-UCRs in patients’ Progression Free Interval.

	Univariate Analysis				
	Variable	HR	Lower 0.95	Upper 0.95	*p*-Value
Overall Breast Cancer	Age	1.00	0.98	1.01	0.7399
	Stage	2.52	1.71	3.73	0.0000 *
	uc.84	1.02	0.62	1.67	0.9349
	uc.138	1.03	0.78	1.36	0.8447
	uc.147	1.00	0.28	3.52	0.9953
	uc.193	1.10	0.78	1.55	0.5714
	uc.268	1.09	0.78	1.51	0.6127
	uc.271	0.91	0.67	1.23	0.5289
	uc.311	0.70	0.32	1.50	0.3530
	uc.376	1.03	0.75	1.42	0.8574
	uc.378	0.87	0.63	1.20	0.3883
	uc.427	0.96	0.68	1.34	0.7956
	uc.456	1.12	0.89	1.41	0.3205
	uc.475	1.07	0.85	1.33	0.5769

* *p*-value < 0.05.

**Table 5 biomolecules-12-00214-t005:** Summary of univariate and multivariate Cox regression analysis of T-UCRs in Progression Free Interval among subtypes.

		Univariate Analysis				Multivariate Analysis			
	Variable	HR	Lower 0.95	Upper 0.95	*p*-Value	HR	Lower 0.95	Upper 0.95	*p*-Value
Luminal A	Age	0.996887	0.975152	1.019106	0.78157				
	Stage	2.105079	1.109962	3.992354	0.019708 *	2.76746	1.4211	5.389	0.00276 *
	uc.84	1.979264	1.037038	3.77757	0.037227 *	0.0687	0.4617	2.485	0.87288
	uc.138	1.542668	0.955461	2.490761	0.075395				
	uc.147	4.894229	1.028127	23.29817	0.045626 *	0.80186	0.3039	16.358	0.43034
	uc.193	1.46312	0.737398	2.903075	0.275879				
	uc.268	1.882928	1.069127	3.316181	0.028443 *	0.84788	1.2133	4.493	0.01112 *
	uc.271	1.022169	0.72409	1.442955	0.900793				
	uc.311	2.114192	0.699488	6.390118	0.184278				
	uc.376	1.663846	0.984975	2.810615	0.056709				
	uc.378	1.062238	0.715155	1.57777	0.764795				
	uc.427	1.508173	0.906657	2.508762	0.113104				
	uc.456	1.839661	1.22984	2.751864	0.003006 *	0.62948	1.1067	3.182	0.01947 *
	uc.475	1.522604	1.042151	2.224555	0.029338 *	0.02543	0.5948	1.769	0.92714
						**HR**	**lower 0.95**	**upper 0.95**	***p*-value**
Luminal B	Age	0.999501	0.968503	1.031491	0.97523				
	Stage	2.052156	0.831668	5.06373	0.111112				
	uc.84	0.156926	0.025691	0.95853	0.04443 *	0.3566	0.060104	2.1159	0.2564
	uc.138	0.735099	0.35422	1.525523	0.408851				
	uc.147	0.002767	4.9 × 10^−5^	0.156281	0.002901 *	0.0134	0.000186	0.9673	0.0482 *
	uc.193	1.501734	0.65642	3.435611	0.33617				
	uc.268	0.670959	0.31922	1.410269	0.292864				
	uc.271	1.032127	0.318954	3.339942	0.957908				
	uc.311	0.046056	0.003497	0.606546	0.017788 *	0.2223	0.014298	3.4558	0.2827
	uc.376	0.609323	0.256861	1.445431	0.260315				
	uc.378	1.3172	0.565736	2.263933	0.72638				
	uc.427	0.540814	0.213756	1.36829	0.191485				
	uc.456	0.671194	0.378525	1.19015	0.171756				
	uc.475	1.136575	0.638469	2.023284	0.663481				
						**HR**	**lower 0.95**	**upper 0.95**	***p*-value**
HER2 positive	Age	1	1	1.047071	0.910025				
	Stage	3.246709	0.970043	10.86666	0.044062 *	4.063	1.07112	15.412	0.03931 *
	uc.84	1.306121	0.257434	6.626739	0.746998				
	uc.138	1.409298	0.761768	2.607252	0.269814				
	uc.147	35.01671	2.190718	559.7114	0.008221 *	66.3233	3.1042	1417.043	0.00725 *
	uc.193	0.738199	0.296394	1.838563	0.514986				
	uc.268	0.576907	0.189042	1.760575	0.333955				
	uc.271	1.067562	0.400901	2.842817	0.895887				
	uc.311	1.396482	0.10022	19.45885	0.803706				
	uc.376	1.940194	0.687189	5.477901	0.207876				
	uc.378	0.072952	0.005262	1.011354	0.047048 *	0.1338	0.01272	1.407	0.09385
	uc.427	2.014339	0.623733	6.505283	0.237475				
	uc.456	1.426328	0.687533	2.959	0.33758				
	uc.475	2.404616	0.916675	6.307773	0.070817				

* *p*-value < 0.05.

## Data Availability

The data presented in this study are available on request from the corresponding author.

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
