# Peer review of "Transcribed Ultraconserved Regions Are Associated with Clinicopathological Features in Breast Cancer"

_biomolecules, 2022, doi:10.3390/biom12020214_

Round 1
Reviewer 1 Report
This manuscript reports results on the expression of Ultraconserved regions (T-UCR) in Breast Cancer, suggesting several associations, with potential clinical interest. It is true that T-UCR are puzzling genomic objects and that very little is known on their meaning in cancer, markedly in Breast Cancer. This is the main reason why this Reviewer considers this paper worth publishing, provided answers to several questions and additional data.
Main points.
One major concern in this study is a relative lack of information on the Brazilian cohort. This is important, since anyone can access and examine TCGA data, while this is novel. Do I understand well that the majority of the tumor tissues examined has just no clinicopathological characterization ? therefore data are available only for 50 samples ? This is a major problem since this cohort can only provide a general comparison with TCGA, but has no value as validation cohort for TCGA-derived observations. Even in this case, it would be essential to show data on this cohort explicitly, as for example with an additional Supplemental table. Also, providing comparison of assayed T-UCRs frequencies with those calculated from TCGA data would add some appreciable value to this study, while at the end of #3.3 Authors just say that “the profile of expression was similar with TCGA data…” in a very unspecified (and useless) manner.
The second point is Statistics. For “combined” ROC curves, M&M reports “binary logistic regression”. More details are required on this point. In addition, all the figures report comparison of values distribution among different groups of samples, e.g. tumors vs normal tissues, or subtypes, or other groups. First, I do not understand why in certain figures the Box-pIot representation is given, whereas in other the rougher point distribution is preferred. Second, which kind of statistical test was used to calculate the probability, was it nonparametric or parametric?
In #3.3, expression of 12 T-UCR analyzed (why only 12 ?) is compared among tumor groups defined by clinicopathological parameters, such as for example, ER/PgR. Since in the incipit we see that IHC was used to stratify, I guess that Authors used the data available in the TCGA database ? or did they try to correlate with mRNA levels derived from TCGA data themselves ?
Finally, Authors have correlated the expression levels of some T-UCR with Overall Survival. From a clinical point of view, it would be even more interesting to see correlations with Disease Free Survival or Distant metastasis free survival. Could Authors comment on this point or, provide novel analysis concerning DFS ?
Minors.
Here and there Authors say that a T-UCR “has prognostic or diagnostic power”. I think that actually it is much more correct speaking about “associations” and “correlation” rather than suggesting diagnostic or prognostic power that, as it is clear, would require prospective validation using a novel, independent BC cohort. The same is valid also for some sentence saying that (end of page 9) “a potential new target in BC”. This can be true for anything else indeed and, as a matter of fact, it is not sustained by any of the proffered results.
Brazilian population has typical multi-ethnic composition. Does any of the reported variants in T-UCR present different frequency as related to ethnies and it is possible to evaluate if any of those has any eQTL ?
Commenting two recently published studies on UC.183, UC.110 and UC.84 as related to miR-221 in Breast Cancer cell lines (Genes 2021) and on UC.51 as promoting proliferation of BC (Clin Exp Metastasis 2021) would possibly give flavour to the Discussion.
Finally, the language requires thorough revision by an English mother tongue colleague. There are several errors and typos, but also unusual oversimplifications.
Author Response
According Review report 1:
“This manuscript reports results on the expression of Ultraconserved regions (T-UCR) in Breast Cancer, suggesting several associations, with potential clinical interest. It is true that T-UCR are puzzling genomic objects and that very little is known on their meaning in cancer, markedly in Breast Cancer. This is the main reason why this Reviewer considers this paper worth publishing, provided answers to several questions and additional data.
Main points.
One major concern in this study is a relative lack of information on the Brazilian cohort. This is important, since anyone can access and examine TCGA data, while this is novel. Do I understand well that the majority of the tumor tissues examined has just no clinicopathological characterization? therefore data are available only for 50 samples ? This is a major problem since this cohort can only provide a general comparison with TCGA, but has no value as validation cohort for TCGA-derived observations. Even in this case, it would be essential to show data on this cohort explicitly, as for example with an additional Supplemental table.
Response: We thank the reviewer for the relevant comment. We totally agree with the importance of complete information about Brazilian cohort.
Following the reviewer's recommendations, we made a great effort with the hospital's clinical members and we were able to complete all the clinical information of the Brazilian cohort. In fact, table 1 was with some outdated data. We sorry about our mistake, we review all data and include information about all 102 samples from Brazilian cohort. So, we updated the table of clinicopathological features from these patients (Table 1).
A new supplementary table 1 with clinical pathological features from Brazilian cohort was added.
“ Also, providing comparison of assayed T-UCRs frequencies with those calculated from TCGA data would add some appreciable value to this study, while at the end of #3.3 Authors just say that “the profile of expression was similar with TCGA data…” in a very unspecified (and useless) manner.”
Response: The frequencies of the 12 T-UCRs analyzed (percentage of detection among analyzed samples) here were calculated to the TCGA and Brazilian cohort. All T-UCRs are expressed in more than 80% of the samples analyzed, as demonstrated in the below table:
|
Frequency (%) |
|
T-UCRs |
TCGA |
Brazilian |
uc.84 |
95.58% |
85% |
uc.138 |
100% |
98.08% |
uc.147 |
87.7% |
92.31% |
uc.193 |
100% |
85% |
uc.268 |
100% |
98.08% |
uc.271 |
83.4% |
96.15% |
uc.311 |
95.34% |
94.23% |
uc.376 |
100% |
98.08% |
uc.378 |
100% |
85% |
uc.427 |
92.7% |
88.1% |
uc.456 |
100% |
98.08% |
uc.475 |
100% |
98.08% |
We included the information that all T-UCRs analyzed had expression detected in more than 80% of the samples analyzed in both cohorts in #3.1 results item (Page 5, line 174-174).
We removed the statement “the profile of expression was similar with TCGA data…” from the #3.3 results.
“The second point is Statistics. For “combined” ROC curves, M&M reports “binary logistic regression”. More details are required on this point.”
Response: Following the suggestions, implemented the description in the methods section (page 4, line 161-167).
“In addition, all the figures report comparison of values distribution among different groups of samples, e.g. tumors vs normal tissues, or subtypes, or other groups. First, I do not understand why in certain figures the Box-pIot representation is given, whereas in other the rougher point distribution is preferred.”
Response: We thank the reviewer for the observation. We standardized all graphics for the box-plot format.
“Second, which kind of statistical test was used to calculate the probability, was it nonparametric or parametric?”
Response: We included this absent information in #2.4. Statistical Analysis item (Page 4, line 149-151). The association between the variables analyzed and T-UCRs expression levels was determined by non-parametric test (Mann-Whitney preceded by Kruskal-Wallis test when multiple comparisons were performed).
“In #3.3, expression of 12 T-UCR analyzed (why only 12?) is compared among tumor groups defined by clinicopathological parameters, such as for example, ER/PgR. Since in the incipit we see that IHC was used to stratify, I guess that Authors used the data available in the TCGA database ? or did they try to correlate with mRNA levels derived from TCGA data themselves ?“
Response: We chose the 12 T-UCRs based on our previous work “A novel lncRNA derived from an ultraconserved region: lnc- uc.147, a potential biomarker in luminal A breast cancer”, where we did the expression analysis of all 481 T-UCRs. And trying to enrich for novel lncRNA T-UCRs, we excluded all T-UCR which overlapped with mature mRNA sequences of protein coding genes. This left us with 125 intronic and 84 intergenic T-UCRs. T-UCRs identified in the majority of the breast cancer samples (>80%) were selected in order to avoid rare transcripts, generating a list of 33 T-UCRs. Analyzing the subgroups of PAM50 gene signature from these 33 T-UCRs, we observed that 12 T-UCRs (uc.84, uc.138, uc.147, uc.193, uc.268, uc.271, uc.311, uc.376, uc.378, uc.427, uc.456, uc.475) had greater difference (p < 0.01) in expression levels among all subtypes.
For TCGA expression analysis, we used clinical data available in TCGA database. IHC data are available (including positivity for ER, PgR and HER receptors, for example). For these markers, we did not used mRNA information. However, for clarification, we added this information in the main text (Page 4, line 134-136).
“Finally, Authors have correlated the expression levels of some T-UCR with Overall Survival. From a clinical point of view, it would be even more interesting to see correlations with Disease Free Survival or Distant metastasis free survival. Could Authors comment on this point or, provide novel analysis concerning DFS?”
Response: We thank the reviewer for this important discussion. Following the suggestions, we looked for disease-free survival follow-up of TCGA data and included information from updated data (Liu et al., 2018. doi: 10.1016/j.cell.2018.02.052).
These data included the four major clinical outcome endpoints and, for less aggressive tumor type like breast cancer data, the paper recommended use of progression-free interval (PFI) and disease-free interval (DFI). Progression-free interval event considered patient having new tumor event whether it was a progression of disease, local recurrence, distant metastasis, new primary tumors in any site, or died with the cancer without new tumor event, including cases with a new tumor event registered in the patient’s medical record but with no site specification. In summary, any change in patient’s cancer status is denoted as an event, but cancer-unrelated deaths are treated as censored cases.
For better describe this analysis, we added a new topic in results: “3.5. T-UCRs Expression in BC Correlates with Patients' Progression Free Interval (PFI)”.
“Minors.
Here and there Authors say that a T-UCR “has prognostic or diagnostic power ”. I think that actually it is much more correct speaking about “associations” and “correlation” rather than suggesting diagnostic or prognostic power that, as it is clear, would require prospective validation using a novel, independent BC cohort. The same is valid also for some sentence saying that (end of page 9) “a potential new target in BC”. This can be true for anything else indeed and, as a matter of fact, it is not sustained by any of the proffered results .”
Response: We thank the reviewer for this observation. We changed the manuscript sentences to use less "prognostic/diagnostic power" and more "association", furthermore, we also changed the title from “Identification of T-UCRs as potential biomarkers in Breast Cancer” to “Transcribed ultraconserved regions are associated with important clinicopathological Features in Breast Cancer” considering reviewer suggestion.
Brazilian population has typical multi-ethnic composition. Does any of the reported variants in T-UCR present different frequency as related to ethnies and it is possible to evaluate if any of those has any eQTL ?
Response: We agree with the reviewer that Brazilian population have a multi-ethinic composition. Particularly, the Brazilian population included in the present study are from a particular hospital in Curitiba-PR, a city with more Caucasian ancestry. Based on self-reported patients’ records about ethnicity, almost 85% were white and 10% black or mixed ancestry. This numbers are in accordance with previous studies describing around 80% of the population from Curitiba self-reported as Euro-descendant (IBGE, 2013), which is in accordance with previous genetic studies (Braun-Prado et al., 2010, Probst et al., 2000).
Furthermore, the T-UCRs are regions extremely conserved among different species, with rare polymorphisms described. We did not find descriptions about variants among the population, so we did not address this field in manuscript.
“Commenting two recently published studies on UC.183, UC.110 and UC.84 as related to miR-221 in Breast Cancer cell lines (Genes 2021) and on UC.51 as promoting proliferation of BC (Clin Exp Metastasis 2021) would possibly give flavour to the Discussion.”
Response: We appreciated the indication of novel works and we added in our manuscript (highlighted line 82, page 02 and lines 348-350 page 14).
“Finally, the language requires thorough revision by an English mother tongue colleague. There are several errors and typos, but also unusual oversimplifications.”
Response: All manuscript had language revision by the Academic Publishing Advisory Center (CAPA) from Federal University of Parana in response to this critique. We included this information in Acknowledgments section.
Reviewer 2 Report
The authors demonstrated some interesting findings between 12 T-UCRs and clinical features of BC. Some suggestion as follows.
In fig.5 and supplemental figures, asterisk should be replaced by the corresponding number and in accordance with the other figures.
Based on their own previous study, the authors evaluated the expression of 12 T-UCRs in the TCGA data and the Brazilian cohort of breast cancer and analyzed the association of with several parameters such as metastasis, age, overall survival and positivity of receptors, which is an original finding and shows a potential clinical value of T-UCRs as a prognostic marker and target for therapies in breast cancer. Meanwhile, the text is clear and easy to read and the conclusions are consistent with the evidence and arguments presented.
Author Response
According Review report 2:
“The authors demonstrated some interesting findings between 12 T-UCRs and clinical features of BC. Some suggestion as follows.
In fig.5 and supplemental figures, asterisk should be replaced by the corresponding number and in accordance with the other figures.
Based on their own previous study, the authors evaluated the expression of 12 T-UCRs in the TCGA data and the Brazilian cohort of breast cancer and analyzed the association of with several parameters such as metastasis, age, overall survival and positivity of receptors, which is an original finding and shows a potential clinical value of T-UCRs as a prognostic marker and target for therapies in breast cancer. Meanwhile, the text is clear and easy to read and the conclusions are consistent with the evidence and arguments presented.”
Response: We thank the reviewer for the observation and comments. We improve the graphs to be in accordance with the other graphs.
Round 2
Reviewer 1 Report
Changes are OK and I think now that all my concersn have been cleared.